# Postpartum management of hypertensive disorders of pregnancy: a systematic review

Alexandra E Cairns,[1] Louise Pealing,[1] James M N Duffy,[1] Nia Roberts,[2] Katherine L Tucker,[1] Paul Leeson,[3] Lucy H MacKillop,[4] Richard J McManus[1]

[1]Nuffield Department of Primary Care Health Sciences, University of Oxford, Oxford, UK
[2]Knowledge Centre, Bodleian Libraries, University of Oxford, Oxford, UK
[3]Cardiovascular Clinical Research Facility, Division of Cardiovascular Medicine, University of Oxford, Oxford, UK
[4]Nuffield Department of Obstetrics and Gynaecology, University of Oxford, Oxford, UK

**Correspondence to**
Dr Alexandra E Cairns;
alexandra.cairns@phc.ox.ac.uk

## ABSTRACT

**Objectives** Hypertensive disorders of pregnancy (HDP) affect one in ten pregnancies and often persist postpartum when complications can occur. We aimed to determine the effectiveness and safety of pharmacological interventions, other interventions and different care models for postpartum hypertension management.

**Design** A systematic review was undertaken. Nine electronic databases, including Medline, were searched from inception to 16 March 2017. After duplicate removal, 4561 records were screened. Two authors independently selected studies, extracted study characteristics and data, and assessed methodological quality.

**Setting** Randomised controlled trials, case–control studies and cohort studies from any country and healthcare setting.

**Participants** Postnatal women with HDP.

**Interventions** Therapeutic intervention for management of hypertension, compared with another intervention, placebo or no intervention.

**Primary and secondary outcome measures** Outcome data were collected for maternal mortality and severe morbidity; systolic, diastolic and mean arterial blood pressure (BP) control; and safety data. Secondary outcome data collected included the length of postnatal hospital stay and laboratory values.

**Results** 39 studies were included (n=2901). Results were heterogeneous in terms of intervention, comparison and outcome requiring a narrative approach. There were insufficient data to recommend any single pharmacological intervention. 18 studies reported calcium-channel blockers, vasodilators and beta-blockers lowered BP postpartum. 12 of these reported safety data. Limited data existed regarding management in the weeks following hospital discharge. Neither loop diuretics (three studies) nor corticosteroids (one study) produced clinical benefit. Uterine curettage significantly reduced BP over the first 48 hours postpartum (range 6–13 mm Hg) compared with standard care (eight studies), with safety data only reported by four of eight studies.

**Conclusion** There was insufficient evidence to recommend a particular BP threshold, agent or model of care, but three classes of antihypertensive appeared variably effective. Further comparative research, including robust safety data, is required. Curettage reduced BP, but without adequate reporting of harms, so it cannot currently be recommended.

## Strengths and limitations of this study

► All types of intervention for the management of postpartum hypertension—medical, surgical and organisation of care—were eligible for inclusion in this review.

► Randomised controlled studies plus other experimental study designs (cohort studies, case–control studies and quasi-randomised studies) were included, and no limitations were imposed in terms of language or publication date, resulting in a comprehensive review.

► This review highlights significant evidence gaps, demonstrating that further comparative research is required, particularly to clarify postpartum antihypertensive selection.

► Although 39 studies were included, the majority had a high risk of bias such that the evidence provided by this review is of low quality.

► The 39 studies reported a broad range of heterogeneous outcomes, limiting meaningful comparison.

## INTRODUCTION

Hypertensive disorders of pregnancy (HDP) often persist following delivery,[1] and sometimes arise de novo postpartum.[2] In both scenarios adverse events can occur during this period. Approximately one-third of eclampsia occurs postpartum, nearly half beyond 48 hours after childbirth.[3–5] Half of the women who sustain an intracerebral haemorrhage in association with pre-eclampsia do so following birth.[6] Women may enter the postnatal period requiring large doses of antihypertensive medication, but the majority will be treatment-free by 3–6 months.[1 7] This rapidly changing blood pressure (BP) poses a challenge in terms of appropriate antihypertensive selection and dose adjustment.

The National Institute for Health and Care Excellence (NICE) recommends frequent postnatal BP monitoring for women with both pre-eclampsia (every 1–2 days for 2 weeks) and gestational hypertension (at

least once between days 3 and 5).[8] The guideline stipulates thresholds for the increase or commencement (≥150/100 mm Hg) and the reduction or cessation (consider <140/90 mm Hg and reduce <130/80 mm Hg) of antihypertensive medication after birth. However, little detail is provided about frequency or proportion of dose reduction or how to manage multiple medications.[8] The American College of Obstetricians and Gynecologists recommends that BP be monitored in hospital (or with an equivalent level of outpatient surveillance) for 72 hours after birth, and checked again 7–10 days postpartum (sooner if a woman is symptomatic).[9] In line with NICE, they propose treating BP when ≥150/100 mm Hg, but add this should be on two measures, 4–6 hours apart. They make no suggestion regarding BP thresholds for medication reduction, implying uncertainty about when to decrease or stop treatment.

A Cochrane review (search date January 2013) evaluated medical interventions for prevention and treatment of postnatal hypertension. This was limited to randomised controlled trials (RCTs) and included only nine studies.[10] Given the paucity of evidence available, due to Cochrane's restriction to randomised trials alone, we have undertaken an updated systematic review of the postpartum management of hypertension in women with HDP with a broader scope, including the full range of interventions studied, and incorporating cohort and case–control studies, alongside RCTs. The following were our specific questions: (1) How should BP be monitored in women with HDP postpartum? (2) What BP thresholds should be used for antihypertensive treatment initiation, adjustment and cessation postpartum? (3) Which antihypertensive medication(s) should be used in postpartum in women with HDP? (4) What are the benefits and harms of other therapeutic interventions for women with HDP postpartum?

## MATERIALS AND METHODS

A protocol, with explicitly defined objectives, study selection criteria, and approaches to assessing study quality, outcomes and statistical methods, was developed (online supplementary appendix S1). This was registered with PROSPERO: International Prospective Register of Systematic Reviews (CRD42015015527) and is available online (http://www.crd.york.ac.uk/PROSPERO/display_record.asp?ID=CRD42015015527). We followed the guidelines for meta-analyses and systematic reviews outlined by the Preferred Reporting Items for Systematic Reviews and Meta-Analyses statement (online supplementary appendix S2).[11]

A systematic literature review was undertaken to capture evidence from human studies regarding postpartum hypertension management in women with HDP, without restriction by language or publication date (online supplementary appendix S1). We searched the following databases, from inception to 16 March 2017: Cochrane Database of Systematic Reviews, Database of Abstracts of Reviews of Effects and Cochrane Central Register of Controlled Trials, Cumulative Index to Nursing and Allied Health Literature, Embase, Medline, PsycINFO, Science Citation Index, Science (Web of Science Core Collection), Social Science Citation Index and Conference Proceedings Citation Index. We hand-searched reference lists and contacted relevant experts for potentially relevant studies, which might have been missed by electronic searches.[12]

We included RCTs, quasi-randomised studies, case–control studies, and prospective and retrospective cohort studies assessing interventions for hypertension management postpartum in women with HDP (gestational hypertension, pre-eclampsia, chronic hypertension and superimposed pre-eclampsia) arising both during pregnancy and de novo in the postnatal period. Consistent with guidance from Cochrane, conference abstracts were included.[5]

Two reviewers (AEC/LP) independently screened the titles and abstracts, and then critically reviewed the full text of selected studies to assess eligibility. Discrepancies were resolved by discussion before independent extraction of relevant data by the two reviewers. For trials with multiple intervention arms, we extracted data from eligible comparison arms. Data were extracted for the primary and secondary outcomes outlined in table 1. Due to the heterogeneous nature of these studies, a narrative synthesis was undertaken.

Two reviewers (AEC/LP) independently assessed each trial's methodological quality using the Cochrane Collaboration's tool for assessing the risk of bias in RCTs,[13] and the Newcastle-Ottawa Scale for case–control and cohort studies.[14] A global assessment of bias across trials was made.

## RESULTS

Our searches yielded 7105 records, and after excluding duplicates 4561 titles and abstracts were screened (figure 1). Eighty full-text articles were assessed: 35 articles were excluded (online supplementary appendix S3). Forty-five articles, representing 39 studies (32 randomised trials, 2 prospective cohort studies and 5 retrospective cohort studies) reporting data from 2901 postnatal participants, met our inclusion criteria (online supplementary appendix S4). Of the 39 studies, 9 (23%) were published only as conference abstracts. No further details were made available following author contact.

A range of interventions were assessed, including antihypertensive medications (18 studies, n=982), loop diuretics (4 studies, n=503), parenteral steroids (1 study, n=157), other medications (6 studies, n=188), uterine curettage (8 studies, n=837) and novel models of care (2 studies, n=234). Of the 39 studies, 9 (23%) included ≥100 participants, and only 2 studies included ≥200 participants.[15 16] Four were from lower middle-income settings[15 17–19] (classified according to the United Nations[20]), and 13/39 (33%) studies had follow-up periods longer than 7 days

**Table 1** Outcome measures

| | Outcome measures | Timing |
|---|---|---|
| Primary outcome(s) | Maternal mortality<br>Maternal morbidity (ischaemic stroke, intracranial haemorrhage, eclamptic seizure; development of pre-eclampsia with severe features; postnatal complication requiring intervention)<br>Systolic blood pressure control<br>Diastolic blood pressure control<br>Mean arterial pressure control<br>Safety data (adverse events or maternal side effects) | Direct maternal deaths up to day 42 postpartum; late maternal deaths up to 1 year postpartum |
| Secondary outcome(s) | Critical care admission<br>Length of hospital stay following delivery<br>Postnatal readmission to secondary care<br>Antihypertensive medication requirement<br>Urine output<br>Laboratory values<br>Other as defined by study | |

(online supplementary appendix S4). Only 5/39 (13%) and 7/39 (18%) studies, respectively, reported maternal mortality or major maternal morbidity, and while the majority of studies did report some measure of BP control, three did not (table 2A,B). Of the 39 studies, 19 (49%) reported safety data (table 2A,B).

Of the 39 studies, 5 (13%) studies (all evaluating antihypertensive medications) involved mixed antenatal and postnatal populations.[17 21–24] Authors were contacted to request their data set for the postnatal participants, but no data were made available. Of the 39 studies, 6 (15%) included participants with chronic hypertension alongside women with de novo HDP (gestational hypertension or pre-eclampsia).[22 23 25–31] Twelve of 39 (31%) included women with eclampsia—in one, all participants were eclamptic (online supplementary appendix S5).[17]

Thirty of 32 (94%) included RCTs were judged to be at high overall risk of bias, by both reviewers, according to the Cochrane tool, 23/32 (72%) for multiple domains. Only 2 of 32 (6%) were thought to be clearly at low risk of bias.[29–32] All included cohort studies were deemed to have a high risk of bias in at least one domain of the Newcastle-Ottawa Scale (online supplementary appendix S6).

### How should BP be monitored postpartum in women with HDP?

No studies specifically addressed the frequency or method of postnatal BP monitoring. Two evaluated the impact of postpartum care organisation (n=234), using the postnatal readmission rate as their primary outcome (online supplementary appendix S4). Neither reported maternal mortality or morbidity, safety data nor any measure of BP control (table 2B).[26 33]

One assessed introduction of a specialised postpartum clinic (no further details were given) and demonstrated an increased postnatal readmission and triage visit rate (22% intervention group, 9% control group: difference 13%, P<0.04), although 86% occurred before a participant was seen in the clinic.[33] The second study evaluated specialist nurse follow-up, including home visits and telephone contact, and reported no significant difference in the postnatal readmission rate compared with standard care.[26]

### What BP thresholds should be used for antihypertensive treatment initiation, adjustment and cessation postpartum?

No relevant studies identified.

### Which antihypertensive medication(s) should be used postpartum in women with HDP?

Fourteen randomised trials (n=645), one quasi-randomised trial (n=15) and three retrospective cohort studies (n=322) evaluated antihypertensive medications (online supplementary appendix S4). Only three studies reported maternal mortality,[29–31 34 35] and three reported maternal morbidity; no differences between groups were reported (table 2A).[29–31 35 36] Twelve studies reported safety data, in comparisons between multiple classes of antihypertensive agents (table 2A); no clear differences were established, although one study found a greater number of minor side effects reported with oral nifedipine than with oral labetalol.[27 28]

The vast majority of included studies evaluated either acute control of severe hypertension (7/18, 39%) or BP control in the few days after delivery and while women remained hospital inpatients (8/18, 44%). Only three studies, two published only as conference abstracts, evaluated BP control in the weeks and months following hospital discharge.[25 27 28 37]

#### Calcium-channel blockers

Three small studies examined oral nifedipine (n=135); nifedipine resulted in a greater decrease in mean arterial pressure (MAP) 18–24 hours after childbirth than placebo (intervention group 93.9±1.6 mm Hg, control group 100.2±2.6 mm Hg: difference 6.3 mm Hg, P<0.05), but not at other time points to 48 hours (one RCT, n=31).[32] Nifedipine controlled severe hypertension to <160/100 mm Hg more quickly than labetalol (intervention group

 

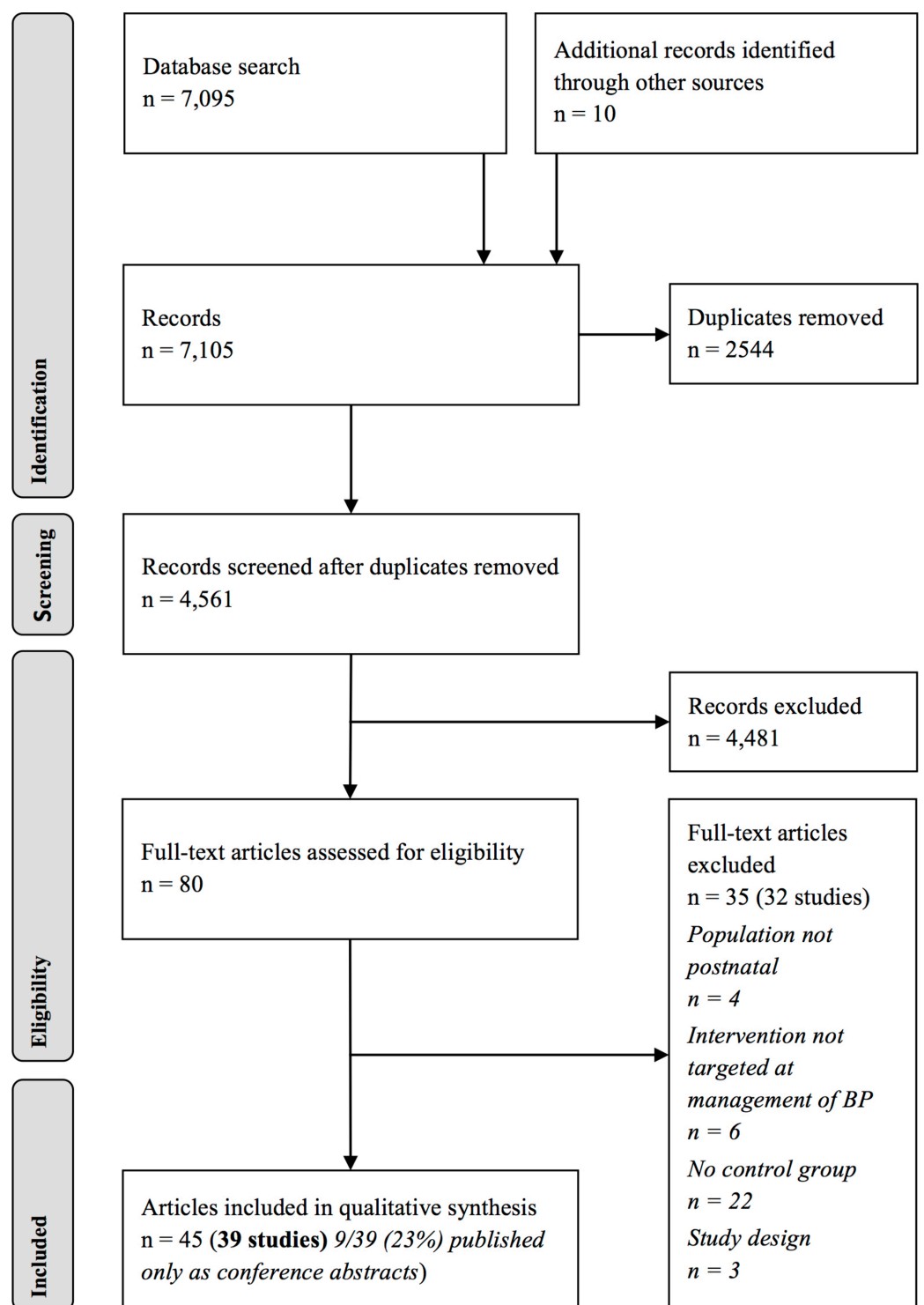

**Figure 1** Preferred Reporting Items for Systematic Reviews and Meta-Analyses flow chart. BP, blood pressure.

25.1±13.6 min, control group 43.6±25.4 min: difference 18.5 min, P=0.002; one RCT, n=21).[21] A single RCT (n=83) reported no significant difference in time taken to control BP to <150/100 mm Hg when comparing nifedipine with methyldopa.[34]

## Vasodilators

Six studies looked at the use of vasodilators (n=252). All used hydralazine via a range of administration routes.

Bolus intravenous hydralazine controlled severe hypertension more quickly than continuous infusion (intervention group 65.23±23.38 min, control group 186.36±79.77 min: difference −121.13 min, P<0.001; one quasi-randomised study, n=15 (postnatal)).[17] Intramuscular hydralazine produced a more significant improvement in MAP at 6 hours than intravenous methyldopa (intervention group 104.5 mm Hg, control group 112 mm Hg: difference −7.5 mm Hg, P=0.0057) but not at other time points

**Table 2** Primary outcome and safety data reporting in included studies

| Study ID | Intervention | Control | Primary outcome assessment | | | | | Safety data reporting | Results (for reported outcomes) |
|---|---|---|---|---|---|---|---|---|---|
| | | | Maternal mortality | Maternal morbidity | SBP control | DBP control | MAP control | | |
| **(A) Antihypertensive medications, 18 studies** | | | | | | | | | |
| **Calcium-channel blockers (three studies)** | | | | | | | | | |
| Barton et al[32] | Nifedipine (oral) | Placebo | | | 🟡 | 🟡 | 🟢 | | SBP control: no significant difference<br>DBP control: no significant difference<br>MAP control: improved in intervention group (difference 6.3 mm Hg, P<0.05) |
| Vermillion et al[21] | Nifedipine (oral) | Labetalol (intravenous bolus) | | | 🟢 | 🟢 | | 🟡 | SBP control: improved in intervention group (difference in time to target BP 18.5 min, P=0.002)<br>DBP control: improved in intervention group (difference in time to target BP 18.5 min, P=0.002)<br>Safety: no significant difference; 1/25 intervention group became hypotensive |
| Sayin et al[34] | Nifedipine (oral) | Methyldopa (oral) | 🟡 | | 🟡 | 🟡 | | | Maternal mortality: no significant difference<br>SBP control: no significant difference<br>DBP control: no significant difference |
| **Vasodilators (6 studies)** | | | | | | | | | |
| Palot et al[36] | Hydralazine (intravenous infusion) plus furosemide (intravenous bolus) | Clonidine (intravenous) plus furosemide (intravenous bolus) | | ⚫ | | | | | Maternal morbidity: no statistical analysis |
| Griffis et al[38 39] | Hydralazine (intramuscular) | Methyldopa (intravenous bolus) | | | | | 🟡 | 🟡 | MAP control: no significant difference<br>Safety: no significant difference; no side effects reported in either group |
| Walss Rodriguez et al[40] | Hydralazine (oral) plus nifedipine (oral, as required) | Nifedipine (oral, as required) | | | 🟡 | 🟡 | | | SBP control: no significant difference<br>DBP control: no significant difference |
| Begum et al[17] | Hydralazine (intravenous infusion) | Hydralazine (intravenous infusion) | | | | 🟢 | | 🟡 | DBP control: improved in intervention group (difference in time to target DBP 121.1 min, P<0.001)<br>Safety: no significant difference; no side effects reported in either group |
| Vigil-De Gracia et al[35] | Hydralazine (intravenous bolus) | Labetalol (intravenous bolus) | 🟡 | 🟡 | 🟡 | 🟡 | | 🟡 | Maternal mortality: no significant difference<br>Maternal morbidity: no significant difference<br>SBP control: no significant difference<br>DBP control: no significant difference<br>Safety: no significant difference; small numbers of side effects reported in both groups |

Continued

**Table 2** Continued

| Study ID | Intervention | Control | Primary outcome assessment | | | | | | Safety data reporting | Results (for reported outcomes) |
|---|---|---|---|---|---|---|---|---|---|---|
| | | | Maternal mortality | Maternal morbidity | SBP control | DBP control | MAP control | | | |
| Hennessy et al[23] | Diazoxide (intravenous bolus) | Hydralazine (intravenous bolus) | | | ● | ● | | | | SBP control: improved in intervention group (difference in percentage achieving target BP 23%, P<0.01) DBP control: improved in intervention group (difference in percentage achieving target BP 23%, P<0.01) |
| **Beta-blockers (5 studies)** | | | | | | | | | | |
| Garden et al[24] | Labetalol (intravenous infusion) | Dihydralazine (intravenous infusion) | | | | ● | | ● | DBP control: no statistical analysis Safety: no statistical analysis; 1/6 intervention group developed bronchospasm; 4/6 control group developed tachycardia and 1/6 developed oliguria; 4/6 control group, drug stopped due to a precipitous fall of DBP to 40–50 mm Hg |
| Fidler et al[42] | Timolol (oral) | Methyldopa (oral) | | | ● | ● | | ● | SBP control: improved in intervention group (difference 5.1 mm Hg, P<0.05) DBP control: no significant difference Safety: no statistical analysis; 1/40 intervention group became disorientated; 1/40 control group became hypotensive and 1/40 became drowsy |
| Mabie et al[22] | Labetalol (intravenous bolus) | Hydralazine (intravenous bolus) | | | | | ● | ● | MAP control: improved in control group (difference 7.8 mm Hg, P=0.02) Safety: no statistical analysis; 1/40 intervention group developed scalp tingling; 2/20 control group developed headaches |
| Shumard et al[41] | Labetalol (oral) | Nifedipine (oral) | | | ● | ● | | | SBP control: improved in control group (difference in time to achieve target BP 1 day, P=0.0031) DBP control: improved in control group (difference in time to achieve target BP 1 day, P=0.0031) |
| Sharma et al[27 28] | Labetalol (oral) | Nifedipine (oral) | | | ● | ● | | ● | SBP: no significant difference DBP: no significant difference Safety: no major side effects reported in either group; minor side effects more commonly reported in control group (20% intervention, 48% control, P=0.04) |
| **Thiazides (2 studies)** | | | | | | | | | | |
| Gaisin et al[25] | Indapamide (oral) | Methyldopa (oral) | | | ● | ● | | ● | SBP control: no significant difference DBP control: no significant difference Safety: no statistical analysis, no details reported |
| Ilshat Gaisin et al[37] | Indapamide (oral) plus ursodeoxycholic acid (oral) | Methyldopa (oral) | | | ● | ● | | ● | SBP control: no significant difference DBP control: no significant difference Safety: no significant difference; no adverse events reported in either group |

Indole alkaloids (1 study)

Continued

**Table 2** Continued

| Study ID | Intervention | Control | Primary outcome assessment | | | | | Safety data reporting | Results (for reported outcomes) |
|---|---|---|---|---|---|---|---|---|---|
| | | | Maternal mortality | Maternal morbidity | SBP control | DBP control | MAP control | | |
| Krebs[43 44] | Reserpine (oral or intramuscular) | Phenobarbital | | | ● | ● | | ● | SBP control: no statistical analysis<br>DBP control: no statistical analysis<br>Safety: no statistical analysis; no adverse events reported in intervention group, no comment on control |
| *Centrally acting alpha agonists (1 study)* | | | | | | | | | |
| Noronha Neto et al[29–31] | Clonidine (oral) | Captopril (oral) | 🟡 | 🟡 | 🟢 | 🟢 | | 🟡 | Maternal mortality: no significant difference<br>Maternal morbidity: no significant difference<br>SBP control: improved in intervention group (difference in number of episodes of high BP (1.4, P<0.08)<br>DBP: improved in intervention group (difference in number of episodes of high BP (1.4, P<0.08)<br>Safety: no significant difference; adverse reactions 18.6% intervention, 28.8% control, P=NS |
| **(B) Loop diuretics, other drugs, uterine curettage and organisation of care, 21 studies** | | | | | | | | | |
| *Loop diuretics (4 studies)* | | | | | | | | | |
| Matthews et al[46] | Furosemide (oral) | Placebo | | | | | 🟡 | | MAP control: no significant difference |
| Ascarelli et al[16] | Furosemide (oral) | No intervention | | 🟡 | 🟡 | 🟡 | | | Maternal morbidity: no significant difference<br>SBP control: no significant difference<br>DBP control: no significant difference |
| Amorim et al[45] | Furosemide (oral) | Placebo | | | 🟢 | 🟢 | 🟢 | | SBP control: improved in intervention group (difference not stated, P<0.001)<br>DBP control: improved in intervention group (difference not stated, P<0.001)<br>MAP control: improved in intervention group (difference not stated, P<0.001) |
| Veena et al[19] | Furosemide (oral)+nifedipine (oral) | Nifedipine (oral) | | 🟡 | 🟡 | 🟡 | 🟡 | | Maternal morbidity: no significant difference<br>SBP control: no significant difference<br>DBP control: no significant difference<br>MAP control: no significant difference |
| *Other drugs (7 studies)* | | | | | | | | | |
| *Selective 5-HT antagonists* | | | | | | | | | |
| Weiner[48] | R41468 (intravenous infusion) | Placebo | | | | | 🟢 | | MAP control: improved in intervention group (difference 25.6mm Hg, P<0.001) |

Continued

**Table 2** Continued

| Study ID | Intervention | Control | Primary outcome assessment | | | | | Safety data reporting | Results (for reported outcomes) |
|---|---|---|---|---|---|---|---|---|---|
| | | | Maternal mortality | Maternal morbidity | SBP control | DBP control | MAP control | | |
| Weiner et al[49] | Ketanserin (intravenous infusion) | Placebo | | | ● | ● | ● | ● | SBP control: improved in intervention group (difference in SBP decline 34 mm Hg, P<0.001) DBP control: improved in intervention group (difference in DBP decline 27 mm Hg, P<0.001) MAP control: improved in intervention group (difference not stated, P<0.001) Safety: no statistical analysis; 3/20 intervention group experienced blurred vision, 1 of these was hypotensive (responded to hydration); 1/20 intervention group experienced mild euphoria |
| Montenegro et al[50] | Ketanserin (intravenous bolus+/− infusion) | Placebo | | | ● | ● | ● | | SBP control: improved in intervention group (absolute difference not stated, P<0.001) DBP control: improved in intervention group (absolute difference not stated, P<0.001) MAP control: improved in intervention group (absolute difference not stated, P<0.001) |
| Alternative therapies | | | | | | | | | |
| Hladunewich et al[51] | L-arginine (oral or intravenous bolus) | Placebo | | | ● | ● | ● | ● | SBP control: no significant difference DBP control: no significant difference MAP control: no significant difference Safety: no significant difference; no adverse events reported in either group |
| Liu et al[52] | Shengkangbao (oral or intravenous bolus) | No intervention | | | ● | ● | | | SBP control: no significant difference DBP control: no significant difference |
| Steroids | | | | | | | | | |
| Barrilleaux et al[53 54] | Dexamethasone (intravenous bolus) | Placebo | | | | | ● | | MAP control: no significant difference |
| Atrial natriuretic peptide | | | | | | | | | |
| Shigemitsu et al[47] | Carperitide (route not specified) | No intervention | ● | | | | ● | ● | Maternal mortality: no significant difference MAP control: no significant difference Safety: no significant difference; no adverse events reported in either group |
| Uterine curettage (8 studies) | | | | | | | | | |
| Salvatore et al[58] | Uterine curettage | No intervention | | ● | ● | ● | | | Maternal morbidity: no statistical analysis SBP control: no statistical analysis DBP control: no statistical analysis |

Continued

**Table 2** Continued

| Study ID | Intervention | Control | Primary outcome assessment | | | | | Safety data reporting | Results (for reported outcomes) |
|---|---|---|---|---|---|---|---|---|---|
| | | | Maternal mortality | Maternal morbidity | SBP control | DBP control | MAP control | | |
| Magann et al[59] | Uterine curettage | No intervention | | | | | 🟢 | 🟡 | MAP control: improved in intervention group (difference at different time points to 24 hours postpartum 6–10 mm Hg, P<0.05) Safety: no significant difference; no complications reported from intervention (follow-up to 7 weeks postpartum) |
| Magann et al[60] | Uterine curettage | Nifedipine (oral) or no intervention | | | | | 🟡 🟢 | 🟡 | MAP control: no significant difference between intervention and oral nifedipine; improved in intervention group compared with no intervention (difference at 8–48 hour postpartum 9–13 mm Hg, P=0.0017) Safety: no significant difference; no complications/side effects reported from interventions (follow-up to 7 weeks postpartum) |
| Gocmen et al[57] | Uterine curettage | No intervention | | | | | 🟢 | | MAP control: improved in intervention group (difference not stated, P=0.01) |
| Gomez et al[61] | Uterine curettage | No intervention | | | | | 🟢 | 🟡 | MAP control: improved in intervention group (difference not stated, P<0.001) Safety: no significant difference; no complications reported from intervention |
| Alkan et al[62] | Uterine curettage | No intervention | | | | | 🟢 | 🟡 | MAP control: improved in intervention group (difference 6.8 mm Hg, P<0.05) Safety: no significant difference; no complications reported from intervention |
| Ragab et al[15] | Uterine curettage | No intervention | 🟡 | ⚫ | | | 🟢 | | Maternal mortality: no significant difference Maternal morbidity: no statistical analysis MAP control: improved in intervention group (difference at 6 hour postpartum 12.3 mm Hg, P=0.02; difference at 24 hours postpartum, 9.2 mm Hg, P=0.01) |
| Mallapur et al[18] | Uterine curettage | No intervention | | | | | 🟢 | | MAP control: improved in intervention group (difference at 4 hour postpartum 7.6 mm Hg, P<0.001). |
| Organisation of care (2 studies) | | | | | | | | | |
| York et al[26] | Nurse specialist follow-up | No intervention | | | | | | | NA |
| Bibbo et al[33] | Specialist postpartum clinic | No intervention | | | | | | | NA |

For primary outcome assessment where there was a significant difference between groups, the magnitude of the difference is reported; where any adverse events or side effects were reported, this is presented.

🟢=improved in intervention group; 🟡=no significant difference; 🔴=improved in control group; ⚫=unclear.

5-HT, 5-hydroxytryptamine; BP, blood pressure; DBP, diastolic blood pressure; MAP, mean arterial pressure; NA, not applicable; NS, non-significant; SBP, systolic blood pressure.

to 24 hours (one RCT, n=26).[38 39] There was no difference in BP control when comparing oral hydralazine with oral nifedipine (one RCT, n=38) or intravenous labetalol (one RCT, n=82).[35 40]

Bolus diazoxide was significantly more effective in achieving a target BP of ≤140/90 mm Hg than intravenous hydralazine (intervention group 67%, control group 43%; relative risk (RR) 0.64, 95% CI 0.46 to 0.89; one RCT, n=37 (postnatal)).[23] One retrospective cohort study did not present any statistical analysis.[36]

### Beta-blockers

Five studies assessed the efficacy of beta-blockers (four RCTs and one retrospective cohort study, n=305). Two RCTs compared intravenous labetalol with intravenous hydralazine/dihydralazine; one involved only six postnatal women and presented no statistical analysis of the data.[24] The other found a significantly greater mean maximal decrease in MAP with intravenous labetalol (intervention group 25.5±11.2 mm Hg, control group 33.3±13.2 mm Hg: difference −7.8 mm Hg, P=0.02; one RCT, n=32 (postnatal)).[22] Results conflicted regarding whether oral labetalol was more or less effective than oral nifedipine; a cohort study reported that labetalol controlled BP less rapidly than nifedipine (intervention group 2.7 days, control group 1.7 days: difference 1.0 days, P=0.0031; one retrospective cohort study, n=128).[41] However, this result was not replicated by an RCT, where the time to BP control was similar in the two groups (n=50).[27 28] Neither study demonstrated a difference in the postnatal length of stay (n=178). Timolol was effective in decreasing diastolic BP on the first day postnatal when compared with methyldopa (intervention group 88.7 mm Hg, control group 93.8 mm Hg: difference −5.1 mm Hg, P<0.05; one RCT, n=80).[42]

### Other antihypertensive medications

No statistically significant difference was found between oral clonidine and oral captopril in the incidence of episodes of severe hypertension postpartum (one RCT, n=90).[29–31] Two RCTs evaluating indapamide versus methyldopa found no difference in BP control over 6–12 months postpartum (n=60).[25 37] One retrospective cohort study (n=140) compared reserpine with phenobarbital; the results suggested that reserpine might achieve faster and greater BP reduction (data extracted from graphs; no statistical analysis). No adverse events were reported in the intervention group.[43 44]

### What are the benefits and harms of other therapeutic interventions for women with HDP postpartum?

### Loop diuretics

Four RCTs (n=503) examined loop diuretics versus placebo or usual care in postpartum hypertension management in women with HDP. None reported maternal mortality or safety data. Only two reported major maternal morbidity, neither demonstrating a difference between groups (table 2B).[16 19]

One RCT (n=120) reported significant improvement in the primary outcome of mean systolic and diastolic BP with oral furosemide versus placebo (magnitude of difference or time points of measurements not stated, P<0.001).[45] This was not the case in the other placebo-controlled RCT, which found no significant difference (n=19).[46] Two further RCTs (n=364) found no significant difference in BP control with oral furosemide versus usual care.[16 19] In one of these, subgroup analysis of women with severe pre-eclampsia (n=70) found women who received oral furosemide had a significantly lower systolic BP at day 2 postpartum (intervention group 142±13 mm Hg, control group 153±19 mm Hg: difference −11 mm Hg, P<0.004), but not at other time points.[16] In the other trial (n=100), furosemide reduced the need for additional antihypertensive treatment during the 3 days of therapy (intervention group 8.0%, control group 26.0%: difference 18%, P=0.017), but this difference did not persist to hospital discharge.[19]

### Other drugs

Five RCTs, one quasi-randomised study and one retrospective cohort study investigated the utility of different drug classes in HDP postpartum (online supplementary appendix S5). Three studies reported safety data, but only one reported maternal mortality, demonstrating no difference between groups,[47] and none reported major maternal morbidity (table 2B).

Three small, crossover RCTs examined the use of selective serotonin receptor inhibitors (SSRIs) compared with placebo (n=55). All studies showed a significant reduction in BP with SSRIs compared with placebo (range 25.6–34 mm Hg).[48–50] These data suggest efficacy for this drug class in hypertension management but do not provide any information regarding relative effectiveness compared with standard antihypertensive drugs. Only one study reported safety data; although no statistical analysis was performed, there were a number of side effects reported in the intervention group.[49]

Two studies evaluated alternative therapies (n=117); there was no difference in BP control with L-arginine supplementation compared with placebo (one RCT, n=45).[51] One reported accelerated recovery of albuminuria with the administration of shengkangbao (Chinese herbal medicine) versus placebo (one quasi-randomised study, n=72). However, the clinical relevance of this outcome is uncertain; there was no difference between groups in the secondary outcomes of systolic BP, diastolic BP or serum creatinine, and no safety data were reported.[52]

A single RCT assessed corticosteroids in the management of severe pre-eclampsia postpartum (n=157).[53 54] No difference was demonstrated between groups in the primary outcome of antihypertensive medication requirement, or in the secondary outcomes of MAP or need for critical care admission, and no safety data were reported. There were small, statistically significant differences found in some laboratory values (platelet count, lactate dehydrogenase and aspartate transaminase). However,

the authors acknowledged that the absolute differences were too small to be clinically relevant.[53]

A very small retrospective cohort study suggested an improvement in MAP with the addition of carperitide (atrial natriuretic peptide) to standard therapy (n=16), and no adverse effects related to the intervention were reported.[47] However, the magnitude of the difference was not published, and the study was too small to draw any firm conclusions.

### Uterine curettage

Six RCTs and two prospective cohort studies (n=837) have explored the role of uterine curettage in post-partum hypertension management. Uterine curettage is a similar process to that used in the surgical management of miscarriage; the lining of the uterus is scraped after completion of the third stage of labour in order to maximise placental tissue removal. This may be under direct vision following caesarean section, or via the transcervical route following vaginal birth. The latter approach may be ultrasound-guided and necessitates some form of anaesthesia. The theory underlying this intervention is that gestational hypertension and pre-eclampsia are placenta-mediated, and therefore ensuring complete evacuation of the uterus following childbirth may accelerate recovery.[55 56]

Seven studies explicitly stated they included both participants who delivered vaginally and those delivered by caesarean; four reported numbers undergoing vaginal delivery (n=248) and caesarean (n=321). One made no comment about the mode of birth.[57] Only one study reported maternal mortality, and there was no difference between groups.[15] Two reported major maternal morbidity, but neither performed any statistical analysis (table 2B). However, both studies did suggest a reduction in the absolute number of eclamptic seizures in the curettage group compared with no intervention.[15 58] In one, however, there was a relevant difference between the study groups; 28/28 (100%) in the control group were eclamptic at enrolment, compared with 9/20 (45%) in the intervention group.[58] Four studies reported safety data, with none reporting any complications related to the intervention (table 2B).[59–62]

All eight studies compared curettage with standard care (ie, no additional intervention), and all suggested that uterine curettage resulted in a significantly lower BP.[15 18 57–62] One of these had two control groups: standard care and oral nifedipine; when compared with oral nifedipine, no difference was noted with curettage.[60]

Five studies reported the magnitude of the difference in MAP between curettage and standard care: range 6–13 mm Hg.[15 18 59 60 62] Only two of these reported BP data beyond 24 hours postpartum; one RCT reported a significantly lower MAP at 48 hours with curettage (intervention group 104 mm Hg, control group 113 mm Hg: difference 9 mm Hg, P=0.0017; n=45),[60] but the other RCT demonstrated no significant difference in MAP at 48 hours (n=420).[15]

One study demonstrated that a greater proportion of the intervention group attained the target BP of <140/90 mm Hg at 24 (intervention group 9/20 (45%), control group 3/28 (11%): difference 34%, no P value quoted) and 48 hours postpartum (intervention group 14/20 (70%), control group 8/28 (29%): difference 41%, no P value quoted).[58] Two studies did not present the size of the difference between groups.[57 61]

## DISCUSSION

This review found evidence demonstrating that calcium-channel blockers, vasodilators and beta-blockers lower BP postpartum, but no clear answer to which was most effective and should, therefore, be preferentially prescribed. All but two studies examined the acute control of severe hypertension or short-term BP control while women remained in hospital postpartum,[25 37] and so provide little guidance about prescription in the weeks after discharge. Moreover these both examined thiazide diuretics, not recommended in the UK for use while breast feeding.[8] Complete safety data were limited across trials, as were data regarding objective clinical outcomes, and two further studies examined antihypertensive agents not recommended for use postpartum in the UK (methyldopa and reserpine).[63 64] One trial evaluated captopril at a much higher daily dose than the UK recommended daily starting dose.[64]

Uterine curettage is not currently recommended, due to safety concerns regarding additional anaesthetic and operative risks, and the availability of alternative treatments to lower BP, particularly in the context of vaginal birth.[65] However, the included studies consistently demonstrated that uterine curettage improved BP control versus standard care,[15 18 57–62] with one reporting an equivalent effect to oral nifedipine.[60] Among the limited safety data, none reported an excess complication rate (infection or uterine damage) with curettage, but given the low incidence of operative complications, the total population (n=837) was likely insufficient to adequately address potential competing risks. Furthermore, these studies did not demonstrate any impact from curettage on maternal mortality or severe morbidity, and concerns exist about some studies' methodology. The evidence reviewed is insufficient to recommend incorporation of this intervention into routine clinical practice.

Four trials evaluating loop diuretics failed to provide conclusive evidence of benefit. Three produced non-significant results in their main analysis,[16 19 46] and the single conference abstract, which did suggest better BP control with oral furosemide, did not publish the magnitude of the difference, rendering it difficult to assess the clinical relevance.[45] In contrast to the Cochrane review, we conclude that, at present, there is no evidence to support the routine use of diuretics postpartum.[10]

We found no adequate evidence to support alternative medications or a particular care model in the management of HDP postpartum. SSRIs substantially reduced BP

versus placebo,[48–50] but no published data were identified comparing their efficacy with standard antihypertensive treatment, making it difficult to draw meaningful conclusions about their clinical application. Neither study evaluating postpartum care organisation reported maternal mortality or morbidity, or any measure of BP control, with both selecting postnatal readmissions as their primary outcome. An increased postnatal readmission rate, however, may not necessarily reflect harm; it might instead suggest that a particular model of care can better detect problems in the community and admit appropriately, ultimately resulting in a lower risk to patients.

In light of the heterogeneous nature of research in this field, when designing this review, we included all interventions targeting hypertension management, but not end-organ complications, including eclampsia. Therefore, trials evaluating magnesium sulfate were outside the scope of this review. We acknowledge the relevance of this therapy in women with severe pre-eclampsia, especially in the immediate postnatal period, and a Cochrane review suggests there is no uncertainty regarding the effectiveness of this therapy.[66]

A strength of this review is that cohort studies, case–control studies and quasi-randomised studies were eligible in addition to RCTs, and no language or date restrictions were imposed, resulting in a comprehensive review that provides evidence suggesting significant research gaps, consistent with the findings from the Cochrane review (2013).[10] The applicability of the findings and recommendations from this review is restricted by the low quality of included studies; both reviewers judged the vast majority to be at high overall risk of bias. Nearly a quarter of the included studies were published only as conference abstracts, and therefore not subjected to peer review. Data extraction was restricted to the information provided in the abstracts (no authors provided additional data on request). These were limiting factors in our analysis, but we nonetheless felt it was important to include these studies for completeness, especially given the paucity of evidence that exists in this field. A further justification for their inclusion is that half of the trials reported in conference abstracts never reach full publication, and positive trials are more likely to be published than negative ones,[67] which has the potential to skew the results of a review if they are omitted.

A further limitation of this review is that the majority of identified studies did not report substantive clinical outcomes such as maternal mortality, morbidity or harms. Without these, it is difficult to define properly the potential role of proposed interventions in clinical practice. The incidence of adverse maternal and neonatal outcomes, particularly in high-resource settings, is low, meaning adequately powering studies for real outcomes of interest is financially demanding. Therefore researchers often employ surrogate outcomes. Additionally, the range of outcomes reported in included studies was broad and inconsistent, with BP changes in particular being measured in a variety of different ways,

further limiting the comparability of trials. Increasingly, core outcome sets are being produced, with a view to trials reporting as standard a minimum set of outcomes that are clinically meaningful and important to patients.[68] We hope in the future that this would enhance our ability to synthesise results from different studies to produce high-quality evidence. There is consensus about trying to move away from surrogate outcomes, for example time to BP control, as they cannot effectively substitute for clinically important outcomes. An important and clinically meaningful end point should measure how a patient feels, functions or survives .

The Cochrane review included only nine randomised trials (author names in bold in online supplementary appendix S4). We believe our review adds to this, as an additional 30 studies are included (19 predating the Cochrane search, and 11 subsequent to it), providing a current and complete summary of all available research in the field. The contrast between the scales of the two reviews highlights a lack of high-quality evidence, despite a reasonably high number of research studies being conducted to answer the question about how hypertension should be managed postpartum in women with HDP. In future, studies need to be more robust and better designed to address the research questions adequately. Furthermore, in spite of these extensions, the body of evidence identified was substantially smaller than that underpinning antenatal hypertension management; 18 studies (n=982), not restricted to RCTs, evaluated antihypertensive medications postpartum. Furthermore, the size of all but a few individual studies was small. In comparison, a Cochrane review (2014) evaluating antihypertensive medication for mild to moderate hypertension in pregnancy included 49 RCTs (n=4723).[69] Moreover, the quantity and quality of evidence supporting the management of HDP are vastly less than that available for essential hypertension outside pregnancy, where individual RCTs commonly involve several thousand participants.[70]

This review demonstrates a lack of good-quality evidence for postpartum hypertension management, emphasising the need for further RCTs directly comparing different antihypertensive agents, BP thresholds for medication adjustment and different models of care, with outcome measures other than postnatal readmissions. We believe the studies examining uterine curettage justify further research to evaluate clinically meaningful outcomes and procedural risks. It might be pragmatic to confine this to curettage at caesarean section, given concerns regarding surgical intervention after vaginal birth; an additional anaesthetic is not required; infection risk is lowered within a sterile surgical field compared with the transcervical route; and curettage under direct vision limits perforation risk. This might be beneficial in women with severe pre-eclampsia, where BP control during pregnancy has been challenging despite multiple medications.[55]

**Acknowledgements** The authors would like to thank Dr Ly-Mee Yu, Dr Helen Cotton and Dr Victoria E Cairns for their assistance with translation.

**Contributors** AEC drafted the protocol with JMND, and drafted and piloted the data extraction sheet. These were reviewed by RJMcM, LP, KLT, LHM and PL. NR and AEC wrote the search strategy, and the online searches were conducted by NR. AEC and LP reviewed the search results independently and carried out the data extraction. This manuscript was drafted by AEC and reviewed by RJMcM, JMND, LP, NR, KLT, LHM and PL. AEC will be the guarantor.

**Funding** The research was funded by the National Institute for Health Research (NIHR) Collaboration for Leadership in Applied Health Research and Care Oxford at Oxford Health NHS Foundation Trust, and via a Research Professorship awarded to RJMcM (NIHR-RP-02-12-015). The views expressed are those of the author(s) and not necessarily those of the NHS, the NIHR or the Department of Health.

**Competing interests** None declared.

**Provenance and peer review** Not commissioned; externally peer reviewed.

**Data sharing statement** Extra data can be accessed via the Dryad data repository at http://datadryad.org/ with the doi:10.5061/dryad.pb6f2.

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
