## [Reviewer comments · BMJ Open]

ARTICLE DETAILS

TITLE (PROVISIONAL)	Postpartum management of hypertensive disorders of pregnancy: a systematic review
AUTHORS	Cairns, Alexandra; Pealing, Louise; Duffy, James; Roberts, Nia; Tucker, Katherine; Leeson, Paul; MacKillop, Lucy; McManus, Richard

VERSION 1 - REVIEW

REVIEWER	Kathryn Sharma Central Coast Perinatal, Santa Barbara, CA USA
REVIEW RETURNED	02-Feb-2017

GENERAL COMMENTS	This article is an attempt to update and summarize the available literature regarding postpartum hypertension management. 1. Thorough abstract, clear methodology, concisely written.2. Minor grammatical and punctuation errors noted throughout paper.3. A paper on management of postpartum hypertension should probably give some discussion or comment to treatment with MgSO₄ for the prevention of seizure. The available literature is quite limited therefore this would not be a difficult task.4. Your discussion is just a reiteration of your results. This should be a place to summarize and draw conclusions regarding your data. This should be revised and greatly truncated.5. Overall, excellent and thorough job. Very much contributes to the available literature.
---

REVIEWER	Tanara Vogel Pinheiro Universidade Federal do Rio Grande do Sul, Brazil
REVIEW RETURNED	07-May-2017

GENERAL COMMENTS	Cairns et al. carried out this interesting and important systematic review surveying the postpartum management of hypertensive disorders of pregnancy. Comparing different interventions is usually a difficult and ambitious task. However, despite the difficulties related to heterogeneity and high risk of bias in many studies, the authors were able to make a very adequate synthesis and to highlight evidence gaps that should be addressed in future studies.
---

REVIEWER	Jennifer Stuart Harvard T.H. Chan School of Public Health and Brigham and Women's Hospital
REVIEW RETURNED	11-May-2017

GENERAL COMMENTS	This article presents a systematic review of management of
--

hypertensive disorders of pregnancy during the postpartum period. The authors sought to update and expand a 2013 Cochrane review by including cohort and case-control studies, in addition to randomized controlled trials. They also sought to expand upon the previous review by evaluating monitoring (in addition to treatment) of blood pressure during this time period but no studies were identified on the frequency or method of blood pressure monitoring postpartum. The authors followed established PRISMA guidelines for systematic reviews and clearly communicated the protocol and article identification strategy. The search resulted in 38 studies included in the review. However, nearly one third of these (n=11, 29%) were published only as conference abstracts and restricted to the information provided in the abstract as the abstract authors did not provide additional data upon request. It is unclear if this prevented proper evaluation of the analyses but inclusion of conference abstracts, which have not undergone peer review, and in the context of abstract authors not providing additional information ultimately raises concern about their inclusion in this review. Reviewed articles evaluated a variety of treatments/interventions but few studies had a tested a consistent intervention, so little consensus could be reached in this review.

The exposure of interest for this review is not clear and is not consistently discussed throughout (“postpartum hypertension”, “postnatal women with hypertensive disorders of pregnancy”, etc.). Did included study participants only have de novo high blood pressure in pregnancy (gestational hypertension and preeclampsia)? What about chronic hypertension alone and/or superimposed on preeclampsia? Postpartum preeclampsia? The manuscript would be improved by clarifying language to make clear the exposure/s of interest. For example, if it was not new onset high blood pressure during postpartum but, rather, new onset high blood pressure arising during pregnancy but also present after delivery perhaps “treatment of hypertensive disorders of pregnancy that persist postpartum” or “postnatal care of women with hypertensive disorders of pregnancy” (or something similar) would be more clear.

The authors note that “the evidence provided by the review is of low quality” and comment on this a number of times. Given the emphasis placed on this and the importance of weighing the quality of included studies, more weight could be given to how the risk of bias was evaluated within each study (summarized in Appendix S6a and S6b), both in the results and the discussion and possibly also the methods (was this determined by the two reviewing authors?). This is especially valuable given that one of the main conclusions is the need for more studies on the topic, so recommendations for how to improve upon previous studies would be beneficial.

Overall, the language could be tightened up a bit and points could be made more succinctly (for example, lines 395-404 could be cut to a single sentence). Appendix S3 is also included within Appendix S1, so consider removing Appendix S3. Appendix S6b could be more clearly presented – alphabetical scale and detailed footnotes make this complex to digest.

While this systematic review article makes the case for more clinical studies of treatment for and management of high blood pressure during postpartum, the variety of outcomes evaluated and lack of consistency across studies prevented consensus and limited the utility of this review. Attempts to improve upon the Cochrane review

	in 2013 were limited by the availability of literature across those domains/research questions.
--	---

REVIEWER	Professor Lucy Chappell King's College London, UK I have co-authored a publication in 2015 with one of the authors, Prof R McManus, on a different topic (predicting pre-eclampsia) but have not worked in this area (postpartum hypertension) with the authors.
REVIEW RETURNED	12-May-2017

GENERAL COMMENTS	This is a well-written, clear manuscript that was easy to read with very good attention to detail. It has originality and would be useful addition to the literature, for a generalist readership. My comments are minor only. 1. Originality The authors have identified the Cochrane review of trials in this area but highlighted the additional studies that they have usefully included. The submitted SR is therefore a more comprehensive overview of the field. 2. Importance of work to general readers This manuscript should be of interest to BMJ Open readers, as the work is relevant to a range of healthcare professionals including those working in primary care, secondary care (obstetrics, general physicians) and midwives and nurses. The Open Access policy will ensure wide availability. 3. Scientific reliability 3a. Research Question: The research questions are clearly identified at the end of the Introduction and appropriately designed for the study. 3b. Overall design of study: The authors have undertaken an appropriate design for this study, registered the protocol prospectively, with no restrictions (e.g. language, date etc.) 3c. Papers/ participants studied: The authors have described the papers and participants clearly in the text and particularly in the Tables. I have one minor comment - it would be useful if Figure 1 could make it clear that of the 42 'articles'/38 studies, 11 were published only as conference abstracts. 3d. Methods: The methods are clearly described and include the PRISMA checklist. I am in agreement that no ethical approval was required. 3e. Results: The results are comprehensive, and I agree that these data are likely to be too heterogeneous for meta-analysis. I have a few minor comments:  - P8, line 202: in the section on antihypertensive treatment the authors could helpfully differentiate between new treatment of acute hypertension, and ongoing treatment of non-acute hypertension (e.g. gestational hypertension or pre-eclampsia
---

	persisting postpartum) although it is possible that this is not clear from the source data. This also relates to the source papers' choice of outcome and its timing as acute treatment trials would typically have a short-term outcome, while ongoing treatment trials are more likely to have a longer term outcome.  - P9, line 211, and others: I think it would be useful if absolute figures could be given as well as the differences to allow readers to assess whether the differences are clinically meaningful. It is possible that this information could be included in the Table (e.g. 2a), rather than the text, but I think useful to have extracted and presented somewhere. For example, the decrease in MAP in Barton 1990 paper is 6.3 mmHg, but this decrease is within the normal range for MAP (93.9 versus 100.2 mmHg) and thus the clinical significance is less clear. By contrast, the difference in time to BP control in the Vermillion 1999 paper (18.5 mins) may be considered clinically significant by some given that the absolute values are 25 +/- 13.6 minutes vs. 43.6 +/- 25.4 minutes; P = .002 whereas if time to control was over 4 hours for both arms, an 18 min difference would not be clinically meaningful. P11, line 270 and others: The authors may wish to rephrase 'the remaining RCT... was negative' as 'did not show a significant difference'. 4. Interpretation and conclusions: The authors clearly present the findings in the wider context of what is known, and highlight the limitations (including the high risk of bias) of the source data. They highlight a clear research gap, in somewhat stark comparison to the extensive literature in the non-pregnant population. I have some minor comments:  - The authors could usefully expand on what they consider to be useful clinical outcomes in this population, (e.g. see p13, line 344) and how this might reflect the difference between acute treatment and ongoing treatment. Although it is difficult to pin down specific important outcomes, the authors could comment on whether time to control, or absolute BP levels might be considered more important, particularly as some aspects of pregnancy hypertension may vary from hypertension in other settings (e.g. balancing the need to avoid severe systolic hypertension against the pros and cons of rapid changes/ falls in blood pressure). The authors could also include a comment on the heterogeneity of outcomes even within a single parameter (e.g. the many ways of measuring and describing changes in BP). - P15, line 396: The authors might wish to change 'significantly' to 'substantially' where they are not describing a statistically tested analysis. - P15, line 423 (minor): 'aesthetic' should read 'anaesthetic' In summary, this is a well-constructed manuscript reporting sufficiently novel findings in a useful format that will be a useful to a range of healthcare professionals. It represents a considerable amount of work and the edits suggested are minor only.
--	--

REVIEWER	Natalie Dayan, Assistant Professor of Medicine McGill University Canada
REVIEW RETURNED	13-May-2017

GENERAL COMMENTS	This systematic review tries to address too many questions. The inclusion of various different designs and outcomes is confusing. The authors should consider narrowing to (a) the most relevant clinical question based on prior/existing knowledge gaps (I would suggest: which anti-hypertensive/drug is more effective at lowering BP/reducing mortality/reducing re-admission (b) include RCT or quasi randomized studies only, and (c) only focus on literature from 2013-present in order to highlight the additional evidence that this review is bringing. The reader is left wondering: so what's new since 2014? This paper does not explicitly tell us this. As it is currently written, it is not clear which are the new studies (other than looking at the years in the tables) or whether this review picked up additional older studies not picked up by the Cochrane review. Limitations are not well discussed.
---

VERSION 1 – AUTHOR RESPONSE

Peer review comment	Response	Section, page and line number
Reviewer #1		
Thorough abstract, clear methodology, concisely written. Overall, excellent and thorough job. Very much contributes to the available literature.	Thank you for your comments.	No action required.
Minor grammatical and punctuation errors noted throughout paper.	Thank you for your comments. All manuscript authors are native English speakers. We have used software including Microsoft Office Spelling and Grammar and Grammarly to assist with identifying and correcting errors.	N/A
A paper on management of postpartum hypertension should probably give some discussion or comment to treatment with MgSO ₄ for the prevention of seizure. The available literature is quite limited therefore this would not be a difficult task.	Thank you for your comments. The review's objective was to evaluate interventions for managing blood pressure. There is no uncertainty regarding the efficacy and safety of magnesium sulphate for the prevention and treatment of eclampsia (Duley et al. 2010). Within the discussion we acknowledge the importance of magnesium sulphate and reference the Cochrane review.	Discussion Page 14 Lines 379-84
Your discussion is just a reiteration of your results. This should be a place to summarize and draw	Thank you for this helpful observation. The discussion has now been	Discussion Page 13

Peer review comment	Response	Section, page and line number
conclusions regarding your data. This should be revised and greatly truncated.	reviewed and updated, including a short summary of the main results, strengths and limitations, and placing results within the wider literature. The total word count of the discussion has been reduced, despite incorporation of the additions suggested by other reviewers. In particular the opening three paragraphs (now split into four paragraphs) have been reduced from 569 words, to 453.	Line 340-437

Reviewer #2

Cairns et al. carried out this interesting and important systematic review surveying the postpartum management of hypertensive disorders of pregnancy. Comparing different interventions is usually a difficult and ambitious task. However, despite the difficulties related to heterogeneity and high risk of bias in many studies, the authors were able to make a very adequate synthesis and to highlight evidence gaps that should be addressed in future studies.

Thank you for your comments.

No action required.

Reviewer #3

This article presents a systematic review of management of hypertensive disorders of pregnancy during the postpartum period. The authors sought to update and expand a 2013 Cochrane review by including cohort and case-control studies, in addition to randomized controlled trials. They also sought to expand upon the previous review by evaluating monitoring (in addition to treatment) of blood pressure during this time period but no studies were identified on the frequency or method of blood pressure monitoring postpartum.

Thank you for your comments.

No action required.

Peer review comment	Response	Section, page and line number
The authors followed established PRISMA guidelines for systematic reviews and clearly communicated the protocol and article identification strategy.		
The search resulted in 38 studies included in the review. However, nearly one third of these (n=11, 29%) were published only as conference abstracts and restricted to the information provided in the abstract as the abstract authors did not provide additional Reviewed articles evaluated a variety of treatments/interventions but few studies had a tested a consistent intervention, so little consensus could be reached in this review. data upon request. It is unclear if this prevented proper evaluation of the analyses but inclusion of conference abstracts, which have not undergone peer review, and in the context of abstract authors not providing additional information ultimately raises concern about their inclusion in this review.	Thank you for your comments. Having updated the search in March 2017, two further full papers have been published, reducing this number to nine. The Cochrane Collaboration recommends including conference abstracts in a systematic review (http://handbook-5-1.cochrane.org/, section 6.2.2.4). The limitations discussed within the reviewer's comments, for example, limited peer review, have been taken into account when assessing the study's risk of bias. We contacted authors seeking clarifications regarding methods and results where necessary. We have described the limitations of including conference abstracts within the discussion. 'Nearly one-quarter of the included studies were published only as conference abstracts, and therefore not subjected to peer review. Data extraction was restricted to the information provided in the abstracts (no authors provided additional data upon request). These were limiting factors in our analysis, but we nonetheless felt it was important to include these studies for completeness, especially given the paucity of evidence that exists in this field. A further justification for their inclusion is that half of the trials reported in conference abstracts never reach full publication, and positive trials are more likely to be published than negative ones,⁶⁶ which has the potential to skew the results of a	Methods Page 6 Lines 141-2 Results Page 7 Lines 158-9 Discussion Pages 14-15 Lines 395-403

Peer review comment	Response	Section, page and line number
The exposure of interest for this review is not clear and is not consistently discussed throughout (“postpartum hypertension”, “postnatal women with hypertensive disorders of pregnancy”, etc.). Did included study participants only have de novo high blood pressure in pregnancy (gestational hypertension and preeclampsia)? What about chronic hypertension alone and/or superimposed on preeclampsia? Postpartum preeclampsia? The manuscript would be improved by clarifying language to make clear the exposure/s of interest. For example, if it was not new onset high blood pressure during postpartum but, rather, new onset high blood pressure arising during pregnancy but also present after delivery perhaps “treatment of hypertensive disorders of pregnancy that persist postpartum” or “postnatal care of women with hypertensive disorders of pregnancy” (or something similar) would be more clear.	review if they are omitted.’ Thank you for your comments. We have explicitly stated the study populations within the results, for example, reporting studies which include women with chronic hypertension or de novo hypertensive disorders of pregnancy (gestational hypertension and pre-eclampsia). ‘6/39 (15%) studies included participants with chronic hypertension alongside women with de novo HDP (gestational hypertension or pre-eclampsia).²² 23 25-31 12/39 (31%) studies included women with eclampsia – in one all participants were eclamptic (Appendix S5).^{17,} Within the introduction we have clarified the breath of participants included within our review. ‘Given the paucity of evidence available, we have undertaken an updated systematic review of the postpartum management of hypertension in women with HDP with a broader scope’	Introduction Page 5 Lines 111-2 Methods Page 6 Lines 128-9 Results Page 7 Lines 173-4
The authors note that “the evidence provided by the review is of low quality” and comment on this a number of times. Given the emphasis placed on this and the importance of weighing the quality of included studies, more weight could be given to how the risk of bias was evaluated within each study (summarized in Appendix S6a and S6b), both in the results and the discussion and possibly also the methods (was this determined by the two reviewing authors? This is especially valuable given that one of the main conclusions is the need for more studies on the topic, so	We have used established tools to assess the risk of bias of the included studies, the Cochrane risk of bias tool was used for randomised studies, and the Newcastle-Ottawa scale for non-randomised studies. ‘Two reviewers (AC/LP) independently assessed each trial’s methodological quality using the Cochrane Collaboration’s tool for assessing the risk of bias in randomised trials,¹³ and the Newcastle-Ottawa scale for case-control and cohort studies.^{14,}’ The appendix provides the authors judgements and justifications.	Methods Page 7 Lines 149-52 Results Pages 7-8 Lines 176-9 Discussion Page 14

Peer review comment	Response	Section, page and line number
recommendations for how to improve upon previous studies would be beneficial.		Line 393-5
Overall, the language could be tightened up a bit and points could be made more succinctly (for example, lines 395-404 could be cut to a single sentence). Appendix S3 is also included within Appendix S1, so consider removing Appendix S3. Appendix S6b could be more clearly presented – alphabetical scale and detailed footnotes make this complex to digest.	Thank you. We have removed Appendix S3. We have edited the discussion section in line with peer review comments. Despite incorporation of the suggested additions, the total word count of the discussion has been reduced, despite incorporation of the additions suggested by other reviewers. In particular the opening three paragraphs (now split into four paragraphs) have been reduced from 569 words, to 453. In Appendix 6b we have added the words Low / Unclear / High with alphabetical scale in brackets to increase clarity.	Discussion Page 14 Line 340 onwards
Reviewed articles evaluated a variety of treatments/interventions but few studies had a tested a consistent intervention, so little consensus could be reached in this review. While this systematic review article makes the case for more clinical studies of treatment for and management of high blood pressure during postpartum, the variety of outcomes evaluated and lack of consistency across studies prevented consensus and limited the utility of this review. Attempts to improve upon the Cochrane review in 2013 were limited by the availability of literature across those domains/research questions.	Thank you for your comments. 'The Cochrane review included only nine trials (author names in bold in Appendix S4). We believe our review adds to this, as an additional 30 studies are included (19 pre-dating the Cochrane search, and 11 subsequent to it), providing a current and complete summary of all available research in the field.'	Discussion Page 14 Lines 388-92
Reviewer #4		
This is a well-written, clear manuscript that was easy to read with very good attention to detail. It has originality and would be useful addition to the literature, for a	Thank you for your comments.	No action required.

Peer review comment	Response	Section, page and line number
generalist readership. My comments are minor only.		
1. Originality The authors have identified the Cochrane review of trials in this area but highlighted the additional studies that they have usefully included. The submitted SR is therefore a more comprehensive overview of the field.	Thank you for your comments.	No action required.
2. Importance of work to general readers This manuscript should be of interest to BMJ Open readers, as the work is relevant to a range of healthcare professionals including those working in primary care, secondary care (obstetrics, general physicians) and midwives and nurses. The Open Access policy will ensure wide availability.	Thank you for your comments.	No action required.
3. Scientific reliability 3a. Research Question: The research questions are clearly identified at the end of the Introduction and appropriately designed for the study. 3b. Overall design of study: The authors have undertaken an appropriate design for this study, registered the protocol prospectively, with no restrictions (e.g. language, date etc.) 3d. Methods: The methods are clearly described and include the PRISMA checklist. I am in agreement that no ethical approval was required. In summary, this is a well-constructed manuscript reporting sufficiently novel findings in a useful format that will be a useful to a range of healthcare professionals. It represents a considerable amount of work and the edits suggested are minor only.	Thank you for your comments.	No action required.

Peer review comment	Response	Section, page and line number
3c. Papers/ participants studied: The authors have described the papers and participants clearly in the text and particularly in the Tables. I have one minor comment - it would be useful if Figure 1 could make it clear that of the 42 'articles'/38 studies, 11 were published only as conference abstracts.	Thank you for your comments. We have adjusted Figure 1 accordingly.	Figure 1 Page 25
3e: Results: The results are comprehensive, and I agree that these data are likely to be too heterogeneous for meta-analysis. I have a few minor comments: P8, line 202: in the section on antihypertensive treatment the authors could helpfully differentiate between new treatment of acute hypertension, and ongoing treatment of non-acute hypertension (e.g. gestational hypertension or pre-eclampsia persisting postpartum) although it is possible that this is not clear from the source data. This also relates to the source papers' choice of outcome and its timing as acute treatment trials would typically have a short-term outcome, while ongoing treatment trials are more likely to have a longer term outcome.	We have included a paragraph summarising whether the antihypertensive treatment trials assessed: acute control of severe hypertension; short term management of postpartum hypertension whilst in hospital following birth; or longer-term management of persisting hypertension following discharge. 'The vast majority of included studies evaluated either acute control of severe hypertension (7/18, 39%), or BP control in the few days after delivery, whilst women remained hospital inpatients (9/18, 50%). Only two studies, both published only as conference abstracts, evaluated BP control in the weeks and months following hospital discharge.'^{25 37,}	Results Page 8 Lines 204-7
P9, line 211, and others: I think it would be useful if absolute figures could be given as well as the differences to allow readers to assess whether the differences are clinically meaningful. It is possible that this information could be included in the Table (e.g. 2a), rather than the text, but I think useful to have extracted and presented somewhere. For example, the decrease in MAP in Barton 1990 paper is 6.3 mmHg, but this decrease is within the normal range for MAP (93.9 versus	We have added the absolute values into the text of the results section. For example: 'Three small studies examined oral nifedipine (n=135): nifedipine resulted in a greater decrease in MAP 18-24 hours after childbirth than placebo (intervention group 93.9±1.6mmHg, control group 100.2±2.6mmHg, difference 6.3mmHg, p<0.05), but not at other time points to 48 hours (one RCT, n=31).'³² Nifedipine controlled severe hypertension to	Results Page 9 Line 211 onwards

Peer review comment	Response	Section, page and line number
100.2 mmHg) and thus the clinical significance is less clear. By contrast, the difference in time to BP control in the Vermillion 1999 paper (18.5 mins) may be considered clinically significant by some given that the absolute values are 25 +/- 13.6 minutes vs. 43.6 +/- 25.4 minutes; P =.002 whereas if time to control was over 4 hours for both arms, an 18 min difference would not be clinically meaningful.	<160/100mmHg more quickly than labetalol (intervention group 25.1±13.6 minutes, control group 43.6±25.4 minutes: difference 18.5 minutes, p=0.002; one RCT, n=21).²¹,	
P11, line 270 and others: The authors may wish to rephrase ‘the remaining RCT... was negative’ as ‘did not show a significant difference’.	Thank you for your comments. We have made the suggested amendments within the manuscript: ‘This was not the case in the other placebo-controlled randomised trial, which found no significant difference (n=19).⁴⁶ Two further RCTs (n=364) found no significant difference in BP control with oral furosemide versus usual care.^{16 19,}	Results Page 10 Lines 265-6
4: Interpretation and conclusions: The authors clearly present the findings in the wider context of what is known, and highlight the limitations (including the high risk of bias) of the source data. They highlight a clear research gap, in somewhat stark comparison to the extensive literature in the non-pregnant population. I have some minor comments: The authors could usefully expand on what they consider to be useful clinical outcomes in this population, (e.g. see p13, line 344) and how this might reflect the difference between acute treatment and ongoing treatment. Although it is difficult to pin down specific important outcomes, the authors could comment on whether time to control, or absolute BP levels might be considered more important, particularly as some	Thank you for the useful comments. We have adjusted this paragraph accordingly: ‘A further limitation of this review is that the majority of identified studies did not report substantive clinical outcomes such as maternal mortality, morbidity or harms. Without these, it is difficult to define properly the potential role of proposed interventions in clinical practice. The incidence of adverse maternal and neonatal outcomes, particularly in high resource settings, is low meaning adequately powering studies for true outcomes of interest is financially demanding. Therefore researchers often employ surrogate outcomes. Additionally, the range of outcomes reported in included studies was broad and inconsistent (Appendix S4), with BP in particular being measured in	Discussion Page 15, Lines 404-18

Peer review comment	Response	Section, page and line number
aspects of pregnancy hypertension may vary from hypertension in other settings (e.g. balancing the need to avoid severe systolic hypertension against the pros and cons of rapid changes/ falls in blood pressure). The authors could also include a comment on the heterogeneity of outcomes even within a single parameter (e.g. the many ways of measuring and describing changes in BP).	a variety of different ways, further limiting the comparability of trials. Increasingly, core-outcome sets are being produced, with a view to trials reporting, as standard, a minimum set of outcomes that are clinically meaningful and important to patients.⁶⁷ We hope in future this would enhance our ability to synthesize results from different studies to produce high-quality evidence. There is consensus about trying to move away from surrogate outcomes, for example time to BP control, as they cannot effectively substitute for clinically important outcomes. An important and clinically meaningful end point should measure how a patient feels, functions, or survives.'	
P15, line 396: The authors might wish to change 'significantly' to 'substantially' where they are not describing a statistically tested analysis.	Change made as suggested.	Discussion Page 14 Line 370
P15, line 423 (minor): 'aesthetic' should read 'anaesthetic'	Change made as suggested.	Discussion Page 16 Line 434
Reviewer #5		
This systematic review tries to address too many questions. The inclusion of various different designs and outcomes is confusing. The authors should consider narrowing to (a) the most relevant clinical question based on prior/existing knowledge gaps (I would suggest: which anti-hypertensive/drug is more effective at lowering BP/reducing mortality/reducing re-admission (b) include RCT or quasi randomized studies only, and (c) only focus on literature from 2013-present in order to highlight the additional evidence that this review is	Thank you for your feedback. After considering this in combination with the other reviews received, we respectfully maintain that inclusiveness is a particular strength of this review. Given the lack of good quality evidence in this field, we believe it is useful at this time to provide a comprehensive review of what has been done to date. This is especially pertinent in light of recommendations from the UK's Chief Medical Officer to increase research into postnatal management, including medically	Tables 2a and 2b Pages 27-31

Peer review comment	Response	Section, page and line number
bringing.	complicated pregnancies. This is an overview review, not just a review of interventions in isolation, as this approach is the most helpful to the end consumer. We believe the review does present the information requested: different interventions are discussed separately within the results, tables, discussion. Randomised trials, quasi randomised trials, and other designs are highlighted within the results, tables, discussion. Publication dates are clearly displayed in the results tables, and 'within-intervention' studies are ordered chronologically.	
The reader is left wondering: so what's new since 2014? This paper does not explicitly tell us this. As it is currently written, it is not clear which are the new studies (other than looking at the years in the tables) or whether this review picked up additional older studies not picked up by the Cochrane review.	We have added a comment in the discussion section to highlight what is subsequent to the Cochrane search (31 Jan 2013), and what were additional, earlier papers, picked up by the broader scope of our protocol: 'The Cochrane review included only nine trials (author names in bold in Appendix S4). We believe our review adds to this, as an additional 30 studies are included (19 pre-dating the Cochrane search, and 11 subsequent to it), providing a current and complete summary of all available research in the field.'	Discussion Page 14 Lines 388-92
Limitations are not well discussed.	We have altered the discussion section such that hopefully it is now clearer where the limitations are being discussed: 'The applicability of the findings and recommendations from this review are restricted by the low quality of included studies: both reviewers judged the vast majority to be at high overall risk of bias (Appendix S6). Nearly one-quarter of the included studies were	Discussion Page 14 Lines 393-426

Peer review comment	Response	Section, page and line number
	published only as conference abstracts, and therefore not subjected to peer review. Data extraction was restricted to the information provided in the abstracts (no authors provided additional data upon request). These were limiting factors in our analysis, but we nonetheless felt it was important to include these studies for completeness, especially given the paucity of evidence that exists in this field. A further justification for their inclusion is that half of the trials reported in conference abstracts never reach full publication, and positive trials are more likely to be published than negative ones,⁶⁶ which has the potential to skew the results of a review if they are omitted. A further limitation of this review is that the majority of identified studies did not report substantive clinical outcomes such as maternal mortality, morbidity or harms. Without these, it is difficult to define properly the potential role of proposed interventions in clinical practice. The incidence of adverse maternal and neonatal outcomes, particularly in high resource settings, is low meaning adequately powering studies for true outcomes of interest is financially demanding. Therefore researchers often employ surrogate outcomes. Additionally, the range of outcomes reported in included studies was broad and inconsistent (Appendix S4), with BP in particular being measured in a variety of different ways, further limiting the comparability of trials. Increasingly, core-outcome sets are being produced, with a view to trials reporting, as standard, a minimum set of outcomes that are clinically meaningful and important	

Peer review comment	Response	Section, page and line number
	to patients.⁶⁷ We hope in future this would enhance our ability to synthesize results from different studies to produce high-quality evidence. There is consensus about trying to move away from surrogate outcomes, for example time to BP control, as they cannot effectively substitute for clinically important outcomes. An important and clinically meaningful end point should measure how a patient feels, functions, or survives. The body of evidence identified was substantially smaller than that underpinning antenatal hypertension management: Eighteen studies (n=982), not restricted to RCTs, evaluated antihypertensive medications postpartum. Furthermore, the size of all but a few individual studies was small. In comparison, a Cochrane review (2014) evaluated antihypertensive medication for mild to moderate hypertension in pregnancy: 49 RCTs were included (n=4,723).⁶⁸ The quantity and quality of evidence supporting the management of HDP is vastly less than that available for essential hypertension outside pregnancy, where individual RCTs commonly involve several thousand participants.⁶⁹	

VERSION 2 – REVIEW

REVIEWER	Jennifer J. Stuart Harvard T.H. Chan School of Public Health, USA
REVIEW RETURNED	03-Aug-2017
GENERAL COMMENTS	This article presents a systematic review of management of hypertensive disorders of pregnancy during the postpartum period (presumably both those persisting after arising during pregnancy and those arising de novo following delivery, although this is not explicitly made clear). The authors sought to update and expand a 2013 Cochrane review by additionally including cohort and case-control studies, in addition to randomized controlled trials. They also

	sought to expand upon the previous review by evaluating monitoring (in addition to treatment) of blood pressure during this time period but no studies were identified on the frequency or method of blood pressure monitoring postpartum. The authors followed established PRISMA guidelines for systematic reviews and clearly communicated the protocol and article identification strategy. The search resulted in 39 studies included in the review. However, nearly one quarter of these (n=9, 23%) were published only as conference abstracts and restricted to the information provided in the abstract, as the abstract authors did not provide additional data upon request. Reviewed articles evaluated a variety of treatments/interventions but few studies had a tested a consistent intervention, so little consensus could be reached in this review. While the authors tout the inclusion of 30 additional studies beyond that included in the 2013 Cochrane review, the quality and heterogeneity of these studies seem to limit the value of this update and expansion of the earlier review, beyond highlighting the need for more studies on postpartum hypertension management. Furthermore, given the reputation of the Cochrane reviews, the fact that 19 of the 30 additional studies pre-dated the Cochrane review (and, as such, were available for the 2013 review) calls into question the appropriateness of their inclusion in the current review.
--	--

REVIEWER	Professor Lucy Chappell King's College London, UK
REVIEW RETURNED	04-Aug-2017
GENERAL COMMENTS	The authors have done a very good job of addressing the referees' comments and I am happy with their revisions.

VERSION 2 – AUTHOR RESPONSE

Reviewer: 1

Reviewer Name: Jennifer J. Stuart

Institution and Country: Harvard T.H. Chan School of Public Health, USA

Competing Interests: None declared

Comment: “This article presents a systematic review of management of hypertensive disorders of pregnancy during the postpartum period (presumably both those persisting after arising during pregnancy and those arising de novo following delivery, although this is not explicitly made clear).”

Response: Thank you for your constructive comment. We have edited the methods section to ensure that this is clear (see below for relevant excerpt).

We included RCTs, quasi-randomised studies, case-control studies, prospective and retrospective cohort studies, assessing interventions for hypertension management postpartum in women with HDP (gestational hypertension, pre-eclampsia, chronic hypertension and super-imposed pre-eclampsia) arising both during pregnancy and de novo in the postnatal period. Consistent with guidance from Cochrane, conference abstracts were included. [Page 6, lines 139-144]

Comment: “The authors sought to update and expand a 2013 Cochrane review by additionally including cohort and case-control studies, in addition to randomized controlled trials. They also sought

to expand upon the previous review by evaluating monitoring (in addition to treatment) of blood pressure during this time period but no studies were identified on the frequency or method of blood pressure monitoring postpartum. The authors followed established PRISMA guidelines for systematic reviews and clearly communicated the protocol and article identification strategy. The search resulted in 39 studies included in the review.”

Response: Thank you.

“However, nearly one quarter of these (n=9, 23%) were published only as conference abstracts and restricted to the information provided in the abstract, as the abstract authors did not provide additional data upon request. Reviewed articles evaluated a variety of treatments/interventions but few studies had a tested a consistent intervention, so little consensus could be reached in this review.”

Response: In the version of the manuscript that was reviewed we had tried to ensure that the rationale for inclusion of conference abstracts was clear, following feedback from the reviewers about the original version.

Comment: “While the authors tout the inclusion of 30 additional studies beyond that included in the 2013 Cochrane review, the quality and heterogeneity of these studies seem to limit the value of this update and expansion of the earlier review, beyond highlighting the need for more studies on postpartum hypertension management. Furthermore, given the reputation of the Cochrane reviews, the fact that 19 of the 30 additional studies pre-dated the Cochrane review (and, as such, were available for the 2013 review) calls into question the appropriateness of their inclusion in the current review.”

Response: We acknowledge the critical role that Cochrane reviews play in summarising the high quality evidence that exists with regard to the ability of RCTs to answer a particular research question. We hope, however, that our review makes clear that its intention was to provide a broad overview of the entire evidence base, as opposed to applying the restrictions inherent to a Cochrane review. There is a notable lack of high quality literature published in this field, so we felt this was a helpful approach as it highlights the research interest in this field, but that in future studies need to be more robust and better designed.

In response to these reviewers’ comments we have adjusted the structure of the discussion section of the manuscript such that comparison with Cochrane review is placed alongside comparison with the antenatal literature (see below for relevant excerpt).

The Cochrane review included only nine randomised trials (author names in bold in Appendix S4). We believe our review adds to this, as an additional 30 studies are included (19 pre-dating the Cochrane search, and 11 subsequent to it), providing a current and complete summary of all available research in the field. The contrast between the scales of the two reviews highlights a lack of high quality evidence, despite a reasonably high number of research studies being conducted to answer the question about how hypertension should be managed postpartum in women with HDP. In future, studies need to be more robust and better designed to address the research questions adequately. Furthermore, in spite of these extensions, the body of evidence identified was substantially smaller than that underpinning antenatal hypertension management: eighteen studies (n=982), not restricted to RCTs, evaluated antihypertensive medications postpartum. Furthermore, the size of all but a few individual studies was small. In comparison, a Cochrane review (2014) evaluating antihypertensive medication for mild to moderate hypertension in pregnancy included 49 RCTs (n=4,723). [Page 15, lines 421-429]

Reviewer: 2

Reviewer Name: Professor Lucy Chappell

Institution and Country: King's College London, UK

Competing Interests: None declared

Comment: "The authors have done a very good job of addressing the referees' comments and I am happy with their revisions."

Response: Thank you very much for this feedback.